# Shallow-Water Hydrothermal Vents as Natural Accelerators of Bacterial Antibiotic Resistance in Marine Coastal Areas

**DOI:** 10.3390/microorganisms10020479

**Published:** 2022-02-21

**Authors:** Erika Arcadi, Eugenio Rastelli, Michael Tangherlini, Carmen Rizzo, Monique Mancuso, Marilena Sanfilippo, Valentina Esposito, Franco Andaloro, Teresa Romeo

**Affiliations:** 1Department of Integrative Marine Ecology, Stazione Zoologica “Anton Dohrn”, Sicily Marine Centre, Contrada Porticatello, 29, 98167 Messina, Italy; monique.mancuso@irbim.cnr.it; 2Department of Marine Biotechnology, Stazione Zoologica “Anton Dohrn”, Fano Marine Centre, Viale Adriatico 1-N, 61032 Fano, Italy; 3Department of Research Infrastructures for Marine Biological Resources, Stazione Zoologica “Anton Dohrn”, Villa Comunale, 80121 Napoli, Italy; michael.tangherlini@szn.it; 4Stazione Zoologica Anton Dohrn–Marine Biotechnology Department, Sicily Marine Centre, Villa Pace, Contrada Porticatello 29, 98167 Messina, Italy; carmen.rizzo@szn.it; 5Institute of Polar Sciences, National Research Council (CNR-ISP), Spianata S. Raineri 86, 98122 Messina, Italy; 6Institute for Marine Biological Resources and Biotechnology (IRBIM), National Research Country (CNR), Messina, Spianata S. Raineri 86, 98122 Messina, Italy; 7Department of Research Infrastructures for Marine Biological Resources, Stazione Zoologica “Anton Dohrn”, Sicily Marine Centre, Contrada Porticatello, 29, 98167 Messina, Italy; marilena.sanfilippo@szn.it; 8Istituto Nazionale di Oceanografia e di Geofisica Sperimentale—OGS Borgo Grotta Gigante 42/C, 34010 Sgonico, Italy; vesposito@inogs.it; 9Department of Integrative Marine Ecology, Stazione Zoologica “Anton Dohrn”, Sicily Marine Centre, Lungomare Cristoforo Colombo (Complesso Roosevelt), 90149 Palermo, Italy; franco.andaloro@szn.it; 10Department of Integrative Marine Ecology, Stazione Zoologica “Anton Dohrn”, Sicily Marine Centre, Via dei Mille 46, 98057 Milazzo, Italy; teresa.romeo@szn.it; 11National Institute for Environmental Protection and Research, Via dei Mille 46, 98057 Milazzo, Italy

**Keywords:** shallow-water hydrothermal vents, antibiotic resistance, heavy metal pollution, marine heterotrophic bacteria, Tyrrhenian sea, Panarea

## Abstract

Environmental contamination by heavy metals (HMs) poses several indirect risks to human health, including the co-spreading of genetic traits conferring resistance to both HMs and antibiotics among micro-organisms. Microbial antibiotic resistance (AR) acquisition is enhanced at sites anthropogenically polluted by HMs, as well as in remote systems naturally enriched in HMs, such as hydrothermal vents in the deep sea. However, to date, the possible role of hydrothermal vents at shallower water depths as hot spots of microbial AR gain and spreading has not been tested, despite the higher potential risks associated with the closer vicinity of such ecosystems to coasts and human activities. In this work, we collected waters and sediments at the Panarea shallow-water hydrothermal vents, testing the presence of culturable marine bacteria and their sensitivity to antibiotics and HMs. All of the bacterial isolates showed resistance to at least one antibiotic and one HM and, most notably, 80% of them displayed multi-AR on average to 12 (min 8, max 15) different antibiotics, as well as multi-HM tolerance. We show that our isolates displayed high similarity (≥99%) to common marine bacteria, affiliating with Actinobacteria, Gammaproteobacteria, Alphaproteobacteria and Firmicutes, and all displayed wide growth ranges for temperature and salinity during in vitro physiological tests. Notably, the analysis of the genomes available in public databases for their closest relatives highlighted the lack of genes for AR, posing new questions on the origin of multi-AR acquisition in this peculiar HM-rich environment. Overall, our results point out that shallow-water hydrothermal vents may contribute to enhance AR acquisition and spreading among common marine bacteria in coastal areas, highlighting this as a focus for future research.

## 1. Introduction

Microbial antibiotic resistance (AR) is spreading critically worldwide, threatening human and animal health [1,2]. Seminal evidence from hospitals and clinical environments has triggered research on AR in natural ecosystems, including soils, glaciers, animals and the ocean [3,4,5,6,7,8,9]. In contrast to most chemical contaminants, AR genes (present both within microbial genomes and in extracellular DNA) are not only persisting, but also multiplying and evolving in the environment by replicating in their hosts and being laterally transferred among different bacteria [10,11]. Notably, AR acquisition and spreading among bacteria follows two main routes, including lateral gene transfer of AR genes and/or genetic mutation and recombination [10,12]. However, a third main route is drastically emerging, linked to the co-selection pressure exerted in the environment by several contaminants, especially heavy metals (HMs) [10,13,14,15,16]. In some cases, both multiple heavy metal resistance (HMR) and AR can be due to the presence of a same mobile genetic element of different genes conferring specific resistances or to a same gene set providing both HMR and AR. In other cases, HMR and AR can result from selection processes in which co-resistance mechanisms are coupled physiologically [17,18,19]. As a result, HM contamination can favour the selection of metal-resistant microbes in the environment, as well as their possibly associated repertoire of AR genes [13,18,20,21]. Although anthropogenic HM pollution is known to favour microbial AR gain and spreading [22,23], little is known about such processes in ecosystems naturally enriched in HMs. In the ocean, hydrothermal systems can be considered pristine habitats typically displaying particularly high HM concentrations [24,25]. Recently, the HM-rich deep-sea Lucky Strike hydrothermal vent at 1700 m water depth in the North Atlantic Ocean has been proposed as a pristine natural hot spot for the acquisition and spreading of genes conferring multiple resistances to HMs and antibiotics [26]. Nevertheless, no information is available to date on the occurrence and patterns of microbial AR in shallow-water hydrothermal vents, although high concentrations of HMs were found in these ecosystems [25,27] and occur worldwide near coastal areas, hence being more closely linked to human health and activities than deep-sea hydrothermal vents [28,29,30]. To start filling this gap, for the present work, we collected waters and sediments at the Panarea shallow-water hydrothermal vents in the Mediterranean Sea [27,31,32], assessing the diversity and testing the sensitivity of culturable marine bacteria to different classes of antibiotics. As a proof of concept, we completed this study by investigating HM concentrations in the collected sediments and concurrent HMR of the identified antibiotic-resistant strains.

## 2. Materials and Methods

### 2.1. Sampling Area and Environmental Characterization

The sampling activities were carried out in two active hydrothermal fields located at depth < 30 m off the Panarea island: (a) Bottaro Crater (38°38′13.50″ N; 15°06′36.00″ E; depth, 11 m); (b) hot/cold vents (38°38′22.97” N; 15°04′45.67” E; depth, 10–12 m) (Figure 1). The SE area off the Bottaro islet experienced a violent gas explosion in the past (between the 2 and 3 November 2002) and is now characterized by a main depression at 11 m depth, 14 m wide and 20 m long [33,34,35,36,37]. The nearshore hot/cold vents (at 10–12 m water depth) are characterized by patches of sediments subjected to very different temperatures, approximately 1 m apart from each other. The “hot” vents are affected by emissions of gas and hot fluids (up to 50 °C on the sediment surface), with low pH values of approximately 5.6 in the sediment porewater and can be recognized by the typical presence of yellow-orange bacterial mats on the sediment surface. Conversely, the “cold” vents are characterized by emissions displaying lower temperatures and typical pH of approximately 7.9 [38,39]. Past investigation of this complex hydrothermal system revealed different fluid origins and processes, influenced by magmatic sources and resulting in substantial enrichment (up to 10,000 times) of many major cations and anions and HMs, including Li, Rb, Fe, Cs, I, Ba, Zn, Pb, As [40]. Recent evidence has shown that such intense water–rock interactions and fluid emissions represent a significant source of HMs in this hydrothermal system, which determine HM enrichment in waters, sediments and local biota [27,41].

For the purpose of this study, samples of sediments and of bottom seawater were collected in September 2019 in each of the targeted hydrothermal sites (Figure 1). The main chemical–physical parameters were measured at all sites through a multi-parametric probe (CTD profiler, SBE 19 Plus SeaCAT probe), pH was determined spectrophotometrically [42] and redox potentials of sediments were measured immediately according to the procedure described by Pearson and Stanley [43]. Surface sediments (top 2 cm) were collected using sterile tubes by one SCUBA diver, while bottom seawater samples were collected by acid-washed Niskin bottles (4 L) and immediately transferred into sterile polycarbonate bottles once retrieved at the surface and then processed as described below for the analyses on bacteria. Heavy metals and metalloids were extracted from the sediment samples according to the EPA 3051 procedure, and then the concentrations of As, Cd, Cr, Hg, Ni, Pb, Cu, Mn, V and Zn were determined by inductively coupled plasma mass spectrometry (ICP-MS) (EPA3051 6020), while the concentrations of Al and Fe were determined by inductively coupled plasma optical emission spectrometry (ICP-OES) (EPA3051 6010).

### 2.2. Isolation and Preliminary Characterization of Bacteria from Sediment and Water Samples

Sediment samples were inoculated in 0.5 g aliquots in 20 mL of Marine Broth 2216, while water samples were enriched 1:1 (vol/vol) with the Marine Broth 2216 culturing medium (Conda, Madrid, Spain). All samples were incubated at 25 °C for two weeks. Appropriate dilutions of such enrichments were then incubated at 25 °C for 7 days on Marine Broth 2216 solid agar to obtain pure bacterial cultures. The obtained bacterial colonies were then characterized in terms of size and shape, texture, and colour. The determination of the Gram reactions of the isolated bacterial strains was performed using the non-staining method described by Buck [44], and morphological observations were carried out by light microscopy (Zeiss Axioscope, 1000×, Carl Zeiss Microscopy GmbH, Jena, Germany).

### 2.3. Antibiotic Susceptibility Tests

All the bacterial strains isolated in pure culture were tested against 17 different antibiotics, selected to span different classes depending on their mechanisms of action.

The tested antimicrobial agents included:

Cell wall inhibitors: (i) Cephalosporins: 1st generation (Cefalexin (CL, 30 μg)), 2nd generation (Cefoxitin (FOX, 30 μg)), 3rd generation (Cefotaxime (CTX, 30 μg)); ii) Beta-lactams, Peptidoglycan inhibitors: Fosfomycin ((FOS, 50 μg)), Oxacillin ((OX, 5 μg)), Penicillin (Penicillin G (P, 10 μg)), Amoxicillin ((AML, 10 μg)) and iii) Glycopeptide antibiotics (Vancomycin, (VA, 30 μg)).

Nucleic acid inhibitors: (i) Quinolones: (Levofloxacin (LEV, 5 μg)), (ii) Fluoroquinolones (Ciprofloxacin, (CIP, 5 μg)), (Flumequine (UB, 5 μg)); (iii) Potentiated sulphonamides (Sulphamethoxazole + Trimethoprim, (SSXT, 25 μg)); (iv) RNA synthesis inhibitors: Rifampicin (Rifampicin, (RD, 30 μg)).

Protein synthesis inhibitors: (i) Lincomycin antibiotics (Clindamycin, (CL, 10 μg)); (ii) Aminoglycoside antibiotics (Gentamicin, (CN, 30 μg)); (iii) Macrolides (Erythromycin, (E, 15 μg)); (iv) Tetracyclines (Tetracycline, (TE, 30 μg)).

The antibacterial tests were performed using the Kirby–Bauer agar disc diffusion method [45], according to the National Committee for Clinical Laboratory Standards [46]. All the antibiotic discs were acquired from Diagnostic Liofilchem. Agar plates were spread with a suspension of each bacterial strain at a concentration of 0.5 McFarland standard; antibiotic discs were placed on the plates, which were then incubated at 24 °C for 24–72 h before assessing the eventual presence of halos of inhibition due to the antibiotic treatment [45]. According to the National Committee for Clinical Laboratory Standards (www.clsi.org, accessed on 14 February 2022), at least two plates, one with no antimicrobial agents and one without bacterial strains, were always included as positive and negative growth controls for each bacterial strain tested for antimicrobial susceptibility. All bacterial strains grew well in their respective positive controls, and negative (sterility) controls always showed no growth, testifying the validity of all tests performed.

### 2.4. Taxonomic Identification of Bacterial Strains and Screening of AR Genes in Phylogenetically Closest Bacteria

The bacterial strains isolated in the present work and found to be multi-resistant to ≥8 antibiotics were taxonomically identified by means of 16S rRNA gene Sanger sequence analysis. To do so, polymerase chain reaction (PCR) rDNA was carried out to amplify the 16S rRNA gene using bacteria universal primers 27F (5′-AGAGTTTGATCCTGGCTCAG-3′) and 1492R (5′-GTTTACCTTGTTACGACTT-3′) [47]. Each PCR was carried out in a 30 μL reaction mixture under the following conditions: initial denaturing step of 5 min at 95 °C; 1 min at 95 °C, 1 min annealing step at 52 °C and 1 min primer extension step at 72 °C for 35 cycles; finally, an extension step of 5 min at 72 °C. The PCR products were cleaned with a Wizard SV Gel and PCR–clean-up system (Promega, Madison, WI, USA) following the manufacturer’s protocol. Sequencing of the amplicons was performed using the BrilliantDye Terminator v3.1 Cycle Sequencing Kit (Nimagen, Nijmegen, The Netherlands), with 27F and 1492R primers, in a 3730 DNA Analyzer (Thermo Fisher, Waltham, MA, USA).

The obtained 16S rRNA gene sequences were quality checked using the Geneious R7 suite [48] to remove low-quality regions and then analysed through the NCBI BLASTn service against the nt database [49] to infer the taxonomic placement of each bacterial strain. For the screening of AR genes in phylogenetically closest bacteria, the obtained 16S rRNA gene sequences were submitted to the SILVA aligner service to retrieve up to 20 neighbouring, phylogenetically related sequences. The bacterial genome names retrieved from the SILVA database based on the accession number of each neighbour were then searched in NCBI GenBank to download the corresponding complete genomes or genome assemblies. These were subsequently used as an input for in silico prediction of potential AR genes with the ABRicate tool (https://github.com/tseemann/abricate on 24 January 2022) using the NCBI reference database provided with the software [50] at 80% of sequence identity and 80% of sequence coverage. Moreover, to complement this screening for eventual information about AR in phylogenetically closest bacteria, we pursued two additional approaches in parallel, either by examining existing literature through Google Scholar (https://www.scholar.google.com; accessed on 14 January 2021) or by searching for closest bacterial relatives within the PATRIC database to assess potential AR phenotypes [51].

### 2.5. Multi-AR Isolates’ In Vitro Physiological Tests

To assess the growth ranges and optima of the isolated strains that displayed multi-AR to ≥8 antibiotics, cultivation tests were conducted at different temperature and salinity conditions. Growth at different temperatures was investigated on cultures for up to 10 days at 4, 15, 25, 40, and 50 °C, while salinity tests were carried out respectively at 30, 50, 100, 150, and 200 g/L NaCl.

### 2.6. Physiological and Genetic Traits for HM Tolerance in Antibiotic-Resistant Bacterial Strains

Metal tolerance of the isolates was determined by the plate diffusion method (Selvin et al., 2009) for the following metals: Hg, Pb, Co, Cu, Zi, As, Cd, Fe at different concentrations (10, 50, 100, 500, 1000, 5000 and 10,000 ppm; standards by Sigma-Aldrich, St. Louis, MI, USA). Aliquots of each metal salt solution in Phosphate Saline Buffer (PBS) (0.5 mL) were added to a central well in each agar plate. Each test was carried out in two replicates, including a treatment with PBS but with no respective metal as negative control. The strains were inoculated in radial streaks in each plate and incubated at 25 °C for 72 h, after which metal tolerance was assessed by measuring the radial growth of each bacterial strain for each treatment, compared with the growth in the respective negative control, and expressed as percentage (50–45.6 mm = high growth, >99%; 45.5–25.6 mm = medium growth, >50 to ≤99%; 25–6 mm = low growth >5% to ≤50%;–and 5–0 mm = no growth, ≤5%). For each HM and for each HM concentration tested, a score of 0, 25, 75 or 100 was assigned, respectively, to strains showing ≤5%, >5% to ≤50%, >50 to ≤99%, or >99% growth compared with their respective HM-free control. Each score was then multiplied for the respective HM concentration tested, and the sum of the scores of the seven concentrations tested was divided by the maximum theoretical score to obtain a metal tolerance index from 0 (= no tolerance) to 1 (= 100% resistance) for each of the bacterial strains.

Moreover, for each identified antibiotic-resistant bacterial strain, PCR assays for genes known to confer metal tolerance were carried out. The list of primers used in this study, with details of gene function, sequence, annealing temperature, amplicon size and detailed references, is reported in Appendix A. Briefly, for each bacterial strain, the DNA was extracted using the MasterPure complete DNA/RNA purification kit (Epicenter, Biotechnologies, Madison, WI, USA) according to the manufacturer’s protocol. Then, to assess the presence/absence of each HMR gene in each extracted DNA, PCR reactions were carried out using the VWR HotStart mix in a final volume of 50 µL, including 2U Taq Polymerase, 0.2 µM of each primer, 0.2 µL dNTP, 50 ng DNA. PCR amplicons were checked by electrophoresis on a 1% agarose gel.

## 3. Results and Discussion

The environmental settings found during our sampling period (September 2019) at the different sites of the Panarea hydrothermal system investigated in this study are reported in Table 1 and confirm evidence from previous surveys in this area [27,33,39]. Differences with values published in some previous studies [35,40] suggest a significant temporal variability of the oceanographic environmental settings in this area, likely profoundly influenced by seasonality and by the ongoing and scattered hydrothermal activity [52].

Overall, in this study, we obtained five bacterial strains isolated from the Bottaro Crater (named BC1 to BC5), five from the “hot” vents (HV1 to HV5), and five from the “cold” vents (CV1 to CV5), for a total of eleven Gram-negative and four Gram-positive strains (Table 2).

The AR assays, conducted on all the bacterial isolates obtained, revealed that all strains were resistant to at least one of the 17 antibiotics tested and, most notably, that 80% of them displayed multi-AR on average to 12 (min 8, max 15) different antibiotics (Figure 2). In particular, strains BC4 and HV3 showed the highest multi-AR, being unaffected by 15 out of the 17 antibiotics tested. Similarly, strains HV5, BC1 and HV1 showed above-average (>12) multi-AR (i.e., 14 for HV5 and 13 for BC1 and HV1) (Figure 2). The highest bacterial AR frequency (i.e., 12 resistant bacterial strains out of 15 strains) resulted in tetracycline and clindamycin (protein synthesis inhibitors). Overall, these results highlight a high frequency of AR across all the antibiotic classes tested (i.e., up to 11 bacterial strains out of 15 resistant-to-cell-wall inhibitors, up to 10 for nucleic-acid inhibitors, up to 12 for protein-synthesis inhibitors) (Figure 2).

Different processes can lead to AR in bacteria and could explain the high frequency of AR found in our study: horizontal gene transfer [53], mutation of antibiotic targets [19] and cross-resistance (a same determinant conferring resistance both to antibiotics and to other compounds, such as heavy metals) [54,55]. The major driver for the development of AR is considered to be the excessive use of antibiotics in humans and animals [56]. However, other factors are known to promote AR. For instance, although the specific mechanisms are still unclear, heavy metals (HMs) have been widely documented to play an important role in the co-selection for AR in bacteria [10,13,14,15,16,18]. The high HM concentrations due to anthropogenic environmental pollution have been extensively documented to promote bacterial AR acquisition and spreading [2,13,14]. Conversely, pristine remote environments naturally enriched in HMs, such as deep-sea hydrothermal vents and subsurface metal mines, have been documented as hot spots of multiple bacterial resistances to HMs and antibiotics [26,57,58]. To the best of our knowledge, our study provides the first evidence of a high frequency of AR among bacteria in the waters and sediments of a shallow-water hydrothermal vent. Considering the high concentrations of HMs documented at our shallow-water hydrothermal sites [27,40,59] and based on similar evidence on HM-rich deep-sea hydrothermal vents [26], our results suggest that HM-rich shallow-water hydrothermal systems can act as natural accelerators of microbial AR in marine coastal environments. Based on current knowledge, this process may be due to the presence of different HMs resistance (HMR) and AR determinants co-located in the same genome (co-resistance) or to a same physiological cellular mechanism that confers both HMR and AR (cross-resistance) [19]. Bacteria are usually sensitive to HMs, but can develop HMR due to genetic mutations or acquisition of novel genetic traits through lateral gene transfer [60,61]. Moreover, mobile genetic elements can host not only multi-HMR and multi-AR determinants, but also mutagenesis induction systems and enhancers of lateral gene transfer, typically expressed by the cell under stressful conditions, which can further enhance HMR and AR acquisition and spreading [62,63]. Our evidence of multi-AR bacteria in the HM-rich shallow-water Panarea hydrothermal system thus opens new perspectives for the understanding of how the high environmental concentrations of HMs can directly or indirectly promote co-selection for AR [18,64,65,66].

In our study, all bacterial isolates that showed multi-AR to ≥8 antibiotics (i.e., 12 out of 15 bacterial strains) were successfully identified as belonging to four classes, mostly Actinobacteria (four strains) and Alphaproteobacteria (four), followed by Gammaproteobacteria (three), and Flavobacteria (one) (Table 3). At the genus level, the strains were affiliated with: *Blastococcus* (one strain), *Alteromonas* (two), *Pseudoalteromonas* (one), *Sulfitobacter* (one), *Muricauda* (one), *Kocuria* (one), *Nocardioides* (one), *Erythrobacter* (three), and *Microbacterium* (one). All of these bacterial taxa have been well documented in coastal environments, either in seawater or marine sediments [67,68,69,70,71,72,73,74,75,76,77,78]. All bacterial isolates showed ≥99% sequence identity to known reference sequences previously deposited in public databases (Table 3), indicating that our cultivation effort succeeded in isolating common bacterial members of the coastal marine bacterial assemblages. All multi-AR bacterial strains obtained showed limited morphological diversity (i.e., either cocci or rods) and tolerance to wide ranges of salinity (most from 30 to 150 g/L NaCl) and temperature (most from 4 to 40 °C) (Table 3). This characterization of our bacterial strains depicts ubiquitous bacterial taxa with common phenotypes (aerobic alkalophilic mesophiles and moderate thermophiles) and confirms previous bacterial cultivation efforts from this area [79], suggesting that eventual HMR and AR determinants possessed by our isolates may be easily available/transferable to coastal marine bacteria in the surroundings of the Panarea hydrothermal system.

The phylogenetic analyses of the bacterial strains isolated in our study, carried out on the SILVA database, showed affiliation to a total of nine bacterial genera (Table 3). The coupling of this phylogenetic classification with genomic search for phylogenetically closest relatives within NCBI GenBank or RefSeq databases allowed for the retrieval of a total of 226 relatives, of which 18 genomes results were available, covering all 9 bacterial genera found in our study (Table 3).

Data mining across the PATRIC database [80] revealed that none of these available phylogenetically closest bacteria displayed AR phenotypes, raising new questions on the origin of the multi-AR phenotypes displayed by the bacterial isolates obtained in our study. The in silico prediction of potential AR genes with the ABRicate tool [50] confirmed the lack of AR genes in the genomes of all closest relatives retrieved for our bacterial isolates, except for one genome (*Erythrobacter seohaensis* SW-135; [81]), which we found to encode a subclass B3 metallo-beta-lactamase. This exception does not contradict our hypothesis. *E. seohaensis* SW-135 has been isolated from a tidal flat of the Yellow Sea in Korea, an ecosystem highly impacted by anthropogenic HM pollution [81,82,83,84,85,86].

However, the genome of the second closest relative retrieved for our *Erythrobacter* strain (*E. litoralis* HTCC2594), isolated from a pristine environment not polluted by HM (i.e., Sargasso Sea surface waters, Atlantic Ocean) [87], was found to encode no AR genes. Such evidence actually supports the hypothesis that HM-rich ecosystems can co-select for bacterial strains that display both HMR and AR, as found in our study. Similarly, one of the strains isolated from the HM-rich deep-sea Lucky Strike hydrothermal vent, recently proposed as a hot spot of HMR and AR acquisition, was also affiliated with *Erythrobacter* and showed both HMR and AR [26]. This suggests that deep-sea and shallow-water hydrothermal ecosystems may share common processes of bacterial AR gain and spreading. Analogously, we found that the closest relative found for our *Sulfitobacter* strain (*Sulfitobacter dubius* KMM 3554T, isolated from an HM-poor natural system in the Japan Sea) [88], was found to encode no AR genes, while *Sulfitobacter dubius* strains isolated from the HM-rich deep-sea Lucky Strike hydrothermal vent displayed multiple HMR and AR [89].

As a proof of concept, we implemented our study with the analysis of the concentrations of HMs in the sediments we collected, as well as with the investigation of the concurrent physiological and genetic traits for HM tolerance in the antibiotic-resistant bacterial strains we identified.

The sediments collected in this study for the isolation of antibiotic-resistant bacteria displayed As concentrations (Appendix A) always exceeding the environmental quality standard (SQA, 12 µg g^−1^) proposed by the Italian Legislative Decree 56/2009, with average values (36.5 µg g^−1^) higher than the concentration above in which adverse biological impacts are always expected (AETs, Apparent Effect Thresholds; [90]). The concentrations of other metals and trace elements were lower (Appendix A) than expected based on current knowledge on the local fluids emissions and in comparison with other hydrothermal vent fields [25,26,27]. The high As concentration coupled with relatively low concentrations of other heavy metals is a feature similar to the Lucky Strike hydrothermal vent sediments proposed as arsenic–antibiotic microbial co-resistance hotspots [26], suggesting that As might play a particularly relevant role in the co-evolution of microbial HMR and AR also at our investigated site. Conversely, as our work was not intended as a detailed investigation of the HM concentrations over the investigated area, we cannot exclude higher HM concentrations in the sediments in the nearby surroundings, possibly due to a high spatial and temporal heterogeneity of the local fluids emissions [25,27].

Notably, the large role of anthropogenic As pollution in enhancing the co-selection and dissemination of AR among microbes has been recently supported by independent experimental and molecular evidence from freshwater, soil and marine aquaculture studies [91,92,93]. Our results thus allow us to hypothesize a possibly relevant role of natural As enrichment in such processes also in benthic ecosystems relatively far from anthropogenic pollution.

Consistent with this hypothesis, our physiological and genetic analyses related to HM tolerance/resistance revealed that As resistance was a conserved trait among our multi-AR bacterial isolates, either in terms of their experimental tolerance to As in agar plate assays, or in terms of PCR amplification of their As-resistance (*arsB*) genes (Figure 3 and Appendix A). At the same time, our results highlight a larger repertoire of physiological and genetic traits related to HM tolerance/resistance for the multi-AR bacterial isolates identified in our study (Figure 3), suggesting that HM and trace elements other than As possibly contribute to the co-selection and spreading of HMR and AR at the investigated site. Our study thus supports current evidence that HMs, beyond their known direct toxic and/or carcinogenic effects [94,95,96,97], can also have an indirect negative role by accelerating AR spreading, which contributes to increase HM potential to hamper environmental and human health [10,13,14,15,16].

Overall, the evidence reported in our study, coupled with recent independent perspectives of co-evolution of HMR and AR [13,14], suggest that such patterns of bacterial acquisition of HMR and AR in HM-rich ecosystems (either anthropogenically polluted or naturally enriched in HMs) are common across different bacterial taxa ubiquitous in marine ecosystems. We thus conclude that shallow-water hydrothermal vents typically understudied compared with their deep-sea counterparts, despite being more easily accessible, widespread and abundant throughout the world’s coastal oceans, can represent a promising target for future research in this filed. Our work suggests that the study of the evolution of the mechanisms underlying the co-selection for HMR and AR in shallow-water-vent bacteria may boost interdisciplinary science advances spanning from microbiology, microbial ecology and ecotoxicology to biochemistry, medicine, and pharmacology. Moreover, we can foresee that shallow-water vents may serve as natural laboratories and/or provide useful bacterial models to better constrain the ecological and mechanistic role of HM pollution on AR acquisition and dissemination in the environment.

## Figures and Tables

**Figure 1 microorganisms-10-00479-f001:**
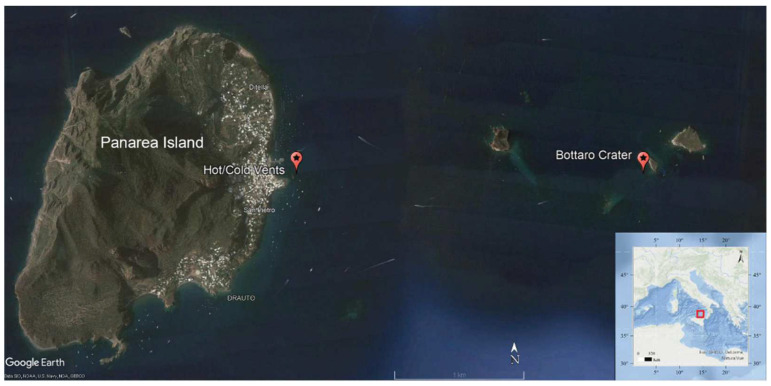
Map of the study area, showing the location of the benthic sites near Panarea Island (Bottaro Crater and hot/cold vents) sampled in the present study (figure generated using Google Earth Pro version 7.3.3.7786, available at https://www.google.com/earth/versions/#earth-pro, accessed on 14 January 2022).

**Figure 2 microorganisms-10-00479-f002:**
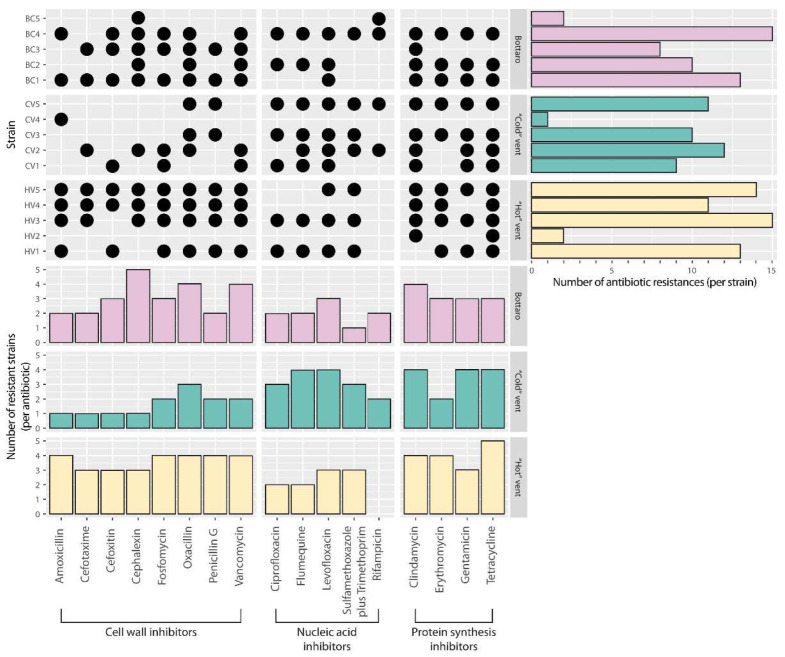
Antibiotic resistance patterns displayed by the bacterial strains isolated in the present study. The black dots highlight the observed resistance of each bacterial strain to a specific antibiotic, while the bar plots show the number of antibiotic resistances found for each of the tested bacterial strains (**right panel**) and the number of resistant bacterial strains for each antibiotic tested (**lower panel**).

**Figure 3 microorganisms-10-00479-f003:**
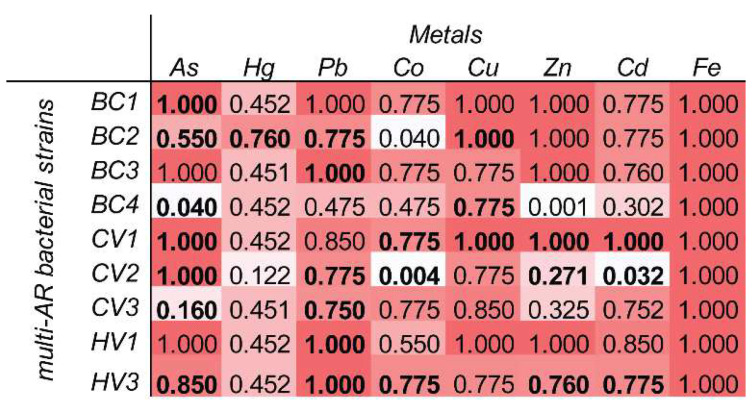
Physiological and genetic traits related to HM tolerance/resistance in the multi-AR bacterial strains isolated in this study. The figure shows the metal-tolerance index values for each of the tested bacterial strains, with white-to-red colour legend to highlight changes in the values from 0 (= no tolerance) to 1 (= 100%, total resistance). In bold, highlighted are the cases in which a specific bacterial strain was positive to the PCR amplification of the resistance gene related to a specific HM or trace element (As resistance, arsB; Co/Zn/Cd resistance, czcA; Hg resistance, merA; Pb resistance, pbrT; Cu resistance, copA). Please note that, unfortunately, the multi-AR strains CV5, HV4 and HV5 could not be assayed for physiological and genetic traits related to HM tolerance/resistance due to failure in further culturing in the lab following AR susceptibility tests.

**Table 1 microorganisms-10-00479-t001:** Environmental settings for bottom waters and sediments at the investigated sites. T, temperature; C, conductivity; ORP, oxygen redox potential; O_2_, oxygen concentration; S, salinity.

Source	Location	Depth (M)	T (°C)	C (ms cm^−1^)	ORP (mV)	O_2_ (mL L^−1^)	pH	S
Bottom water	Bottaro Crater	7.5	19.26	50.46	-	5.48	8.17	37.75
	“Hot” vent	11.41	25.39	57.75	-	4.75	8.07	38.17
	“Cold” vent	11	26.2	58.47	-	4.71	7.89	38.02
Sediment	Bottaro Crater	7.5	28	-	106.27	-	6.78	-
	“Hot” vent	11.7	48	-	8.07	-	5.54	-
	“Cold” vent	11	26	-	7.87	-	5.91	-

**Table 2 microorganisms-10-00479-t002:** Origin and characteristics of the bacterial strains isolated in the present study.

Strain	Origin	Cell Shape	Gram Staining (+/−)	Colony Phenotype (Colour/Aspect)
**BC1**	Bottaro Crater	cocci	-	white/creamy
**BC2**	Bottaro Crater	rods	-	yellow/opalescent
**BC3**	Bottaro Crater	cocci	+	white/pearl
**BC4**	Bottaro Crater	cocci	+	orange-pink/brilliant
**BC5**	Bottaro Crater	rods	-	yellow/brilliant
**CV1**	“Cold” vent	cocci	+	yellow/brilliant
**CV2**	“Cold” vent	rods	+	yellow-white/creamy
**CV3**	“Cold” vent	cocci	-	orange/brilliant
**CV4**	“Cold” vent	cocci	-	Brown/liquid
**CV5**	“Cold” vent	rods	-	orange/creamy
**HV1**	“Hot” vent	rods	-	white/creamy
**HV2**	“Hot” vent	cocci	-	yellow/creamy
**HV3**	“Hot” vent	rods	-	white/creamy
**HV4**	“Hot” vent	rods	-	yellow-white/creamy
**HV5**	“Hot” vent	rods	-	yellow/brilliant

**Table 3 microorganisms-10-00479-t003:** Taxonomic analysis based on 16S rRNA gene sequencing of isolated multi-AR bacterial strains. Reported is the phylogeny for each identified bacterial strain that showed AR to ≥8 antibiotics, based on the results of BLASTn searches against the NT database of the obtained 16S rRNA gene sequences obtained in this study. Reported are also the salinity and temperature ranges determined for each multi-AR bacterial strain during laboratory cultivation tests conducted in this study.

Strain	Class	Family	Ref Strain	Identity	S Range	T Range
**BC1**	Gammaproteobacteria	*Aeromonadaceae*	*Alteromonas marina* strain ROA053	100%	30–150	15–40 °C
**BC2**	Alphaproteobacteria	*Sphingomonadaceae*	*Erythrobacter pelagi* strain UST081027-248	99.8%	30–150	4–40 °C
**BC3**	Actinobacteria	*Microbacteriaceae*	*Microbacterium invictum* strain DC-200	99.7%	30–150	15–40 °C
**BC4**	Actinobacteria	*Geodermathophiliaceae*	*Blastococcus deserti* strain SYSU D8006	99.3%	30–100	15–50 °C
**CV1**	Actinobacteria	*Micrococcaceae*	*Kocuria* sp. strain HY2	100%	30–100	4–40 °C
**CV2**	Actinobacteria	*Nocardioidaceae*	*Nocardioides marinus* strain CL-DD14	99.5%	30–150	15–40 °C
**CV3**	Alphaproteobacteria	*Sphingomonadaceae*	*Erythrobacter pelagi* strain UST081027-248	99.0%	30–50	4–40 °C
**CV5**	Alphaproteobacteria	*Sphingomonadaceae*	*Erythrobacter sp.* strain S44	99.8%	30–100	4–40 °C
**HV1**	Gammaproteobacteria	*Aeromonadaceae*	*Alteromonas macleodii* strain NBRC 102226	99.7%	30–150	15–40 °C
**HV3**	Gammaproteobacteria	*Aeromonadaceae*	*Pseudoalteromonas shioyasakiensis* strain SE3	99.8%	30–150	15–40 °C
**HV4**	Alphaproteobacteria	*Rhodobacteraceae*	*Sulfitobacter faviae* strain S5-53	99.8%	30–150	4–40 °C
**HV5**	Firmicutes	*Flavobacteriaceae*	*Muricauda taeanensis* strain 105	99.8%	30–100	15–50 °C

## Data Availability

The datasets presented in this study have been deposited in online repositories. The name of the repository and accession numbers can be found at https://figshare.com/s/29f9567641b8658036e9, accessed on 14 January 2022.

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
