# Peer review of "Shallow-Water Hydrothermal Vents as Natural Accelerators of Bacterial Antibiotic Resistance in Marine Coastal Areas"

_microorganisms, 2022, doi:10.3390/microorganisms10020479_

Round 1

Reviewer 1 Report

Knowledge of the mechanisms of resistance and the methods of their acquisition by bacteria is extremely important in the era of increasing antibiotic resistance and common occurrence of resistance determinants. Heavy metals, which pollute the environment, can be a selection factor that promotes the spread of antibiotic resistance by co-selection.

The reviewed work deals with this phenomenon, in addition, the authors deal with the antibiotic resistance of cultured marine microorganisms isolated from specific environments such as shallow-water hydrothermal vents. This proves the originality of the research.

It is a very interesting publication properly planned and executed. Methods of analysis are appropriate and recommended from international organizations. It is worth emphasizing that there is a clear description of the research results along with the discussion.

The results obtained by the authors are valuable and draw attention to the presence of the phenomenon of co-selection among microorganisms consisting in the simultaneous selection of genetic determinants of antibiotic resistance in the presence of genes that determine heavy metal resistance. In addition, the authors note the high concentration of As in the studied environment and suggest that the  might play a particularly relevant role in the co-evolution of microbial resistance to metals and antibiotics. Interestingly, they also draw attention to the possibility of obtaining phenotypic resistance by bacteria.

So the authors should be congratulated on their very good work.

There is one issue that should be described in more detail. It is a method of sampling. They were collected in difficult conditions. Has a method of standardization in their collection been adopted?

Author Response

We thank Rev#1 for her/his praise and comments on our work, as well as for the request to specify the method used for sample collection. Actually, the sampling was not difficult, as the sampling site is a coastal area and water temperatures were high, but not precluding manual sampling by scuba divers (so, maximum reliability and possibility to visually select the desired sampling spot). As written in the methods section of our amended manuscript: “Surface sediments (top 2 cm) were manually collected using sterile tubes by one SCUBA diver, while bottom seawater samples were collected by acid-washed Niskin bottles (4 L), immediately transferred into sterile polycarbonate bottles once retrieved at the surface”.

Reviewer 2 Report

The aim of this work was to assess the diversity of culturable marine bacteria and their sensitivity to different classes of antibiotics and heavy metals. Hydrothermal vents at shallower water depths, for which no data are available, were studied in contrast to hydrothermal vents in the deep sea. The results suggest that hydrothermal vents at shallow water depths may contribute to the acquisition and spread of microbial antibiotic resistance among common marine bacteria in coastal areas.
The article makes an important contribution to research on the effects of heavy metals on the environment. It is professionally and correctly written, with clear tables and pictures.
My only suggestion is that the authors describe the major health effects of heavy metals on human health since heavy metal pollution is the focus of the article.

Author Response

We thank Rev#2 for her/his very positive comments on our work, and for the request to add a description of the major health effects of heavy metals on human health.

Accordingly, we have implemented the discussion about this point by highlighting the potential toxic and/or carcinogenic activity of several heavy metals, also citing appropriate references on this issue (including a relevant book and detailed review papers on this topic. See page 11, and added references 94-97).